# Synthetic Model Combination: An Instance-wise Approach to Unsupervised Ensemble Learning

**Alex J. Chan**
University of Cambridge
Cambridge, UK
ajc340cam.ac.uk

**Mihaela van der Schaar**
University of Cambridge
Cambridge Centre for AI in Medicine
Cambridge, UK
mv472@cam.ac.uk

## Abstract

Consider making a prediction over new test data without any opportunity to learn from a training set of labelled data - instead given access to a set of expert models and their predictions alongside some limited information about the dataset used to train them. In scenarios from finance to the medical sciences, and even consumer practice, stakeholders have developed models on private data they either cannot, or do not want to, share. Given the value and legislation surrounding personal information, it is not surprising that only the models, and not the data, will be released - the pertinent question becoming: how best to use these models? Previous work has focused on global model selection or ensembling, with the result of a single final model across the feature space. Machine learning models perform notoriously poorly on data outside their training domain however, and so we argue that when ensembling models the weightings for individual instances must reflect their respective domains - in other words models that are more likely to have seen information on that instance should have more attention paid to them. We introduce a method for such an instance-wise ensembling of models, including a novel representation learning step for handling sparse high-dimensional domains. Finally, we demonstrate the need and generalisability of our method on classical machine learning tasks as well as highlighting a real world use case in the pharmacological setting of vancomycin precision dosing.

## 1 Introduction

Sharing data is often a very problematic affair - before we even arrive at whether a stakeholder will want to, given the modern day value - it may not even be allowed. In particular, when the data contains identifiable and personal information it may be inappropriate, and illegal, to do so. A common solution is to provide some proxy of the true data in the form of a generated fully synthetic dataset (Alaa et al., 2020) or one that has undergone some privatisation or anonymisation process (Elliot et al., 2018; Chan et al., 2021), but this can often lead to low-quality or extremely noisy data (Alaa et al., 2021) that is hard to gain insight on. Alternatively, groups may release models that they have trained on their private data. If for a given task there are multiple of such models, we are left with the task of how to best use these models in combination when they have potentially conflicting predictions. This problem is known as unsupervised ensemble learning since we have no explicit signal on the task, and we aim to construct a combination of the models for making future predictions (Jaffe et al., 2016).

Without having trained the models ourselves, it is a very challenging task for us to know how well the individual models should perform, making the task of choosing the most appropriate model (or ensemble) difficult. This is compounded by the problem that the provided models could perform poorly for not one, but two main reasons: Firstly, the model itself may not have been flexible

enough to properly capture the underlying true function present in the data; and secondly, in the area that they are making a prediction there may not have been sufficient training data used for the model to have been able to learn appropriately - i.e. the model is extrapolating (potentially unreasonably) to cover a new feature point - the main issue in covariate shifted problems (Bickel et al., 2009). Current practice is usually to try and select the *globally optimal* model (Shaham et al., 2016; Dror et al., 2017), that is to say of those made available, which model should be used to make predictions for any given test point in the feature space. This approach potentially addresses the first point, it completely overlooks the second - and this is what we will focus on. In order to consider this problem of extrapolation amongst the individual models, the important question is: what might a solution look like? We must consider the *desiderata* that given no augmentation of the individual models, the ensemble weights must vary depending on the test features and additionally, these weights should reflect the confidence that a model will be able to make an appropriate prediction, and we should be able to tell generally when our confidence is low.

**Contributions** In this work we make a three-fold contribution. First, we establish and document the need for instance-wise predictions in the setting of unsupervised ensemble learning, in doing so introducing the concept of Synthetic Model Combination (SMC), shown in Figure 1. Second, we introduce a novel unsupervised representation learning procedure that can be incorporated into SMC - allowing for more appropriate separation of models and estimation of ensemble weights in sparse high-dimensional settings. Finally, we provide practical demonstrations of both the success and failure cases of traditional methods and SMC, in synthetic examples as well as a real example of precision dosing - code for which is made available at `https://github.com/XanderJC/synthetic-model-combination`, along with the larger lab group codebase at `https://github.com/vanderschaarlab/synthetic-model-combination`.

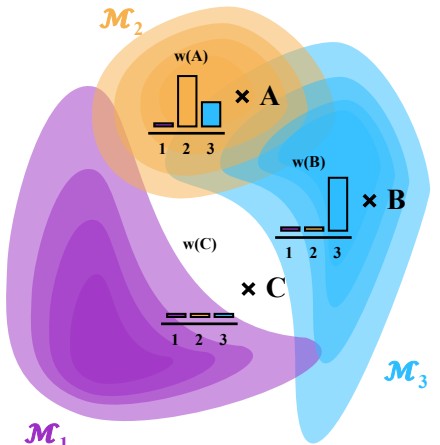

Figure 1: **Instance-wise Ensembles**. Here we represent the density of the training features for three separate models - $\mathcal{M}_1$, $\mathcal{M}_2$, and $\mathcal{M}_3$. Given new test points A, B, and C, we need to construct predictions from these models. A is well represented by both $\mathcal{M}_2$ and $\mathcal{M}_3$ while B only has significant density under $\mathcal{M}_3$. C looks like none of the models will be able to make confident predictions.

## 2 Background

**Formulation** Consider having access only to $N$ provided tuples $\{(\mathcal{M}_j, \mathcal{I}_j)\}_{j=1}^N$ of models $\mathcal{M}_j$ and associated information $\mathcal{I}_j$. With each $\mathcal{M}_j : \mathcal{X} \mapsto \mathcal{Y}$ being a mapping from some covariate space $\mathcal{X}$ to another target space $\mathcal{Y}$ and having been trained on some dataset $\mathcal{D}_j$ which is not observed by us although is in some way summarised by $\mathcal{I}_j$. We are then presented with some other test dataset $\mathcal{D}_T = \{x_i\}_{i=1}^M$ and consequently tasked with making predictions.

**Goal** Our aim is to construct a convex combination of models in a way so as to produce optimal model predictions $\hat{y} = \mathcal{M}^*(x) = \sum_{j=1}^N w(x)_j \mathcal{M}_j(x)$ with $\sum_{j=1}^N w(x)_j = 1$ and $w(x)_j$ taking the $j$th index of the output of the function $w : \mathcal{X} \mapsto \Delta^N$ that maps features to a set of weights on the probability $N$-simplex. We can summarise our task process as:

**Given:** $\{(\mathcal{M}_j, \mathcal{I}_j)\}_{j=1}^N$ and $\{x_i\}_{i=1}^M$    **Obtain:** $w$ to predict $\{\hat{y}_i\}_{i=1}^M$

Having established the setting we are focused on, we can explore contemporary methods and compare how they relate to our work. We consider methods that are interested ultimately in the test-time distribution of labels conditional on feature covariates $p_{(Y|X)}^{test}$. We discuss the differences in methods focuses on the *distributional information available to them*, summarised in Table 1. For example, in standard supervised learning, samples of the feature-label joint distribution $p_{(X,Y)}^{train}$ are given - the training-time conditional distribution is then estimated and assumed to be equivalent at test-time.

Table 1: **(Un-)Related Fields** and how they compare in terms of: A) whether they make instance-wise predictions; B) the distributional information which they require; C) the form of the information; and D) the target quantity they are focused on. References given as: [1] Torrey and Shavlik (2010); [2] Raftery et al. (1997); [3] Ren et al. (2019); [4] Ruta and Gabrys (2005); [5] Jaffe et al. (2016).

| Problem | Ref. | A) Instance-wise | B) Distribution | C) Information | D) Target |
|---|---|---|---|---|---|
| Transfer Learning | [1] | ✓ | $p_{(X,Y)}^{train}, p_{(X,Y)}^{test}$ | $\mathcal{D}_{Train}$ | $p_{(Y|X)}^{test}$ |
| Bayesian Model Averaging | [2] | ✗ | $p_{(X,Y)}^{test}$ | $\mathcal{D}_{Val}$ | $p(\mathcal{M}_i|\mathcal{D})$ |
| Out-of-distribution Detection | [3] | ✗/✓ | $p_{(X)}^{test}$ | - | $p_X^{train}(x^{test}) < \epsilon$ |
| Majority Voting | [4] | ✗ | $p_{(X,Y)}^{train}$ | - | $p_{(Y|X)}^{test}$ |
| Unsupervised Ensemble Regression | [5] | ✗ | $p_{(Y)}^{test}$ | $\mathbb{E}[Y], Var[Y]$ | $\hat{w}$ |
| Synthetic Model Combination | **[Us]** | ✓ | $\{p_{(X)}^{\mathcal{M}_j}\}_{j=1}^N$ | $\{\mathcal{I}_j\}_{j=1}^N$ | $w(x)$ |

In our case, we assume information from considerably less informative distributions, the training time feature distributions $\{p_{(X)}^{\mathcal{M}_j}\}_{j=1}^N$, were the information can take a variety of forms, most practically though through samples of the features or details of the first and second moments. Practically, this appears to be the minimal set of information for which we can do something useful since if only given a set of models and no accompanying information then there is no way to determine which models may be best in general let alone for specific features.

**How are models normally ensembled?** The literature on ensemble methods is vast, and we do not intend to provide a survey, a number of which already exist (Sagi and Rokach, 2018; Dong et al., 2020). The focus is often on training ones own set of models that can then be ensembled for epistemic uncertainty insight (Rahaman et al., 2021) or boosted for performance (Chen and Guestrin, 2016).

In terms of methods of ensembling models that are provided to a practitioner (instead of ones trained by them as well) then closest is the setting of *unsupervised ensemble regression* - which like us does not consider any joint feature label distribution. To make progress though some information needs to be provided, with Dror et al. (2017) considering the marginal label distribution $p_{(Y)}^{test}$, making the strong assumption the first two moments of the response are known. Instead of being directly provided the mean and variance, another strand of work assumes conditional independence given the label (Dawid and Skene, 1979), meaning that any predictors that agree consistently will be more likely to be accurate. Platanios et al. (2014) and Jaffe et al. (2016) relax this assumption through the use of graphical models and meta-learner construction respectively, with a Bayesian approach proposed by Platanios et al. (2016) Recent work of Shaham et al. (2016) attempts to learn the dependence structure using restricted Boltzmann machines.

**Model averaging when validation data is available.** Moving from the unsupervised methods mentioned above, which at most considered information from only marginal distributions, we come to the case of when we may have some validation data from the joint distribution $p_{(X,Y)}^{test}$ (or one assumed to be the same). This can effectively be used to evaluate a ranking on the given models for more informed ensembles. This ensemble can be created in a number of ways (Huang et al., 2009), including work that moves in the direction of instance-wise predictions by dividing the space into regions before calculating per-region weights (Verikas et al., 1999). A practical and more common approach is Bayesian Model Averaging (BMA) (Raftery et al., 1997). Given an appropriate prior, we calculate the posterior probability that a given model *is the optimal one* - and once this is obtained the models can be marginalised out during test time predictions, creating an ensemble weighted by each model's posterior probability. The posterior being intractable, the probability is approximated using the Bayesian Information Criterion (BIC) (Neath and Cavanaugh, 2012) - which requires a likelihood estimate over some validation set and is estimated as: $p(\mathcal{M}_i|\mathcal{D}) = exp(-\frac{1}{2}BIC(\mathcal{M}_i))/\sum_{i=1}^N exp(-\frac{1}{2}BIC(\mathcal{M}_i))$ With this, along with all the ensemble methods previously mentioned, it is important to note the subtle difference in setup to the problem we are trying to work with. In all cases, it is assumed that there is some ordering for the models that holds across the feature space and so a *global* ensemble is produced with a fixed weighting $\hat{w}$ such that $w(x) = \hat{w} \ \forall x \in \mathcal{X}$. This causes failure cases when there is variation in the models across the feature space, since it is a key point that BMA is not model combination (Minka, 2000). This being an important distinction and one of the main reasons BMA has been shown to perform badly under covariate shifted tasks (Izmailov et al., 2021). That being said, it can be extended by

considering the set of models being averaged to be every possible combination of the provided models (Kim and Ghahramani, 2012), although this becomes even more computationally infeasible.

**Is this some form of unsupervised domain adaptation or transfer learning then?** Given the focus on models performing on some region of the feature space outside their training domain this may seem like a natural question. Unsupervised domain adaptation represents this task at an individual model level but usually considers, access to unlabelled data in the target domain as well as labelled data from a (different) source domain $p_{(X,Y)}^{train}, p_{(X)}^{test}$ (Chan et al., 2020). We refer the interested reader to Kouw and Loog (2019) for a detailed review given space constraints.

In a very similar vein, the transfer learning (Torrey and Shavlik, 2010) task also involves a change in feature distribution but instead of being completely unsupervised tends to include some labels on the target set, thus requiring some information on the joint distribution at both test and training time $p_{(X,Y)}^{train}, p_{(X,Y)}^{test}$. A great deal of work then involves learning a prior on the first domain that can be updated appropriately on the target domain (Raina et al., 2006; Karbalayghareh et al., 2018). In contrast to both of these areas, we do not aim to improve the performance of the individual models, but rather combine them based on how well we expect them to perform in the new domain.

## 3    Introducing Synthetic Model Combination

Our success hinges on the assumption that *the quality of a model's prediction will depend on the context features*. That is to say that the performance ordering of the models will not stay constant across the feature space. With this being the case, it should seem obvious that the weightings of individual models should depend on the features presented. As such, we introduce our notion of Synthetic Model Combination[1] - a method that constructs a representation space within which we can practically reason, enabling it to select weights for the models based on a given feature's location within the representation space. Recalling our starting point of $\{(\mathcal{M}_j, \mathcal{I}_j)\}_{j=1}^N$ and $\{x_i\}_{i=1}^M$, we proceed in three main steps which are outlined below:

1. Estimate densities for models: From each information $\mathcal{I}_j$, we generate a density $p_j^{\mathcal{X}}(x)$.

2. Learn low-dimensional representation space: Using $\{x_i\}_{i=1}^M$ and $\{p_j^{\mathcal{X}}(x)\}_{j=1}^N$, learn a mapping to a low dimensional representation space $f_\theta : \mathcal{X} \mapsto \mathcal{Z}$.

3. Calculate ensemble weights for predictions: Evaluate weights $w(x)$ so that predictions can then be made as $\hat{y} = \sum_{j=1}^N w(x)_j \mathcal{M}_j(x)$.

These steps are highlighted again in the full Algorithm 1 as well as shown pictorially in Figure 2.

### 3.1    From Information to Probability Densities

The first step in SMC is to use the information $\mathcal{I}_j$ to produce a density estimate such that we can sample from each model's effective support. Given the flexibility in the form of what we allow $\mathcal{I}_j$ to take, SMC must remain relatively agnostic to this step. A common example of the type of information we expect will simply be example feature samples, and in this case a simple kernel density estimate (Terrell and Scott, 1992) or other density estimation method could be employed. On the other hand, in the medical setting for example, when models are published authors will often also provide demographics information on the patients that were involved in the study, such as the mean and variance of each covariate recorded. In this case we may simply want to approximate the density using a Gaussian and moment-matching for example. When the information is provided in the form of samples it is possible to skip this step, as they can be used directly in the subsequent representation learning step's losses.

### 3.2    Learning a Separable and Informative Space

The point of learning a new representation space - and not simply using the original feature space - is twofold. Firstly, we would like to *reduce the dimensionality*, leading to a more compact rep-

---

[1]Our name is a nod towards synthetic control (Abadie et al., 2010), as we construct a new *synthetic model* as a convex *combination* of others that we think will be most appropriate for a given instance.

resentation and is important because in higher dimensions the densities we model will often end up effectively non-zero in only very small regions, making the last step of SMC difficult. Secondly, we would like to *induce structure*, so that distances and regions in the space better reflect how capable different models will be over the space. This should aid in better selection of model weights when making predictions by allowing SMC to better understand the relationship between the models and the representation space. We should note that if the feature dimensions are low then this step is not *strictly* necessary - the model densities may have appropriate coverage of the space, but this does not then allow you to obtain none of the benefits of the second point.

To proceed we define a space $\mathcal{Z}$ on which we will work and aim to learn a parameterised mapping $f_\theta : \mathcal{X} \mapsto \mathcal{Z}$ such that the representations of features on this space is useful and aids us in our goal of constructing instance-wise ensembles. In each optimisation step of our algorithm we will sample a model dataset $\mathcal{D}_M = \{\hat{x}_j\}_{j=1}^N$ - sampling one feature example [2] from each density associated with a model. This model dataset will play an important part in the learning process as these examples serve as proxies to the regions in the feature space that a model is confident on. Now, given both $\mathcal{D}_T$ and $\mathcal{D}_M$, we aim to construct an optimisation target for a representation learning step that can be learnt end-to-end

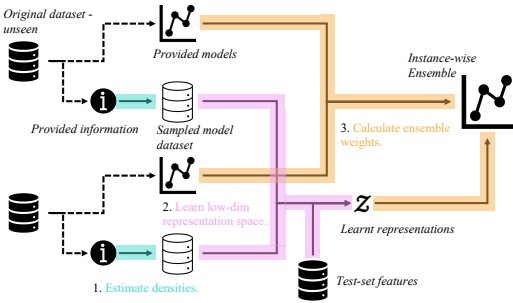

Figure 2: **Diagrammatic Model Approach.** We visually represent the steps (coloured) and objects (*italics*) involved in the SMC approach.

in an autoencoder fashion. As such, we introduce a total loss that can be broken down into three parts:

$$\mathcal{L}_{total} = \mathcal{L}_{reconstruction} + \mathcal{L}_{connection} + \mathcal{L}_{seperation}. \tag{1}$$

The first, $\mathcal{L}_{reconstruction}$, is the simplest as a simple regularised autoencoder loss and is used so that the latent space can accurately represent and reconstruct the feature space:

$$\mathcal{L}_{rec} = \sum_{x_i \in \mathcal{D}_T \cup \mathcal{D}_M} ||g_\phi(f_\theta(x_i)) - x_i||^2 + \beta||f_\theta(x_i)||^2. \tag{2}$$

This introduces a second function $g_\phi : \mathcal{Z} \mapsto \mathcal{X}$ that can be jointly optimised in order to learn the space using standard autoencoder techniques. Alternatively, to incorporate a probabilistic element, we can substitute in a ($\beta$-)VAE loss (Kingma and Welling, 2013; Higgins et al., 2016) by considering samples from an approximate posterior, the probabilistic nature has been shown to improve interpretability in the representation space (Burgess et al., 2018).

We next introduce the first of our more specialised components with the $\mathcal{L}_{connection}$ loss:

$$\mathcal{L}_{con} = \sum_{(\hat{x}_i, \hat{x}_j) \in \mathcal{D}_M \times \mathcal{D}_M} \underbrace{\left(1 - D_{\mathcal{Y}}(\mathcal{M}_i(\hat{x}_i)||\mathcal{M}_j(\hat{x}_i))\right)}_{\text{Predictive Similarity Score}} \times ||f_\theta(\hat{x}_i) - f_\theta(\hat{x}_j)||^2, \tag{3}$$

which is designed so that models that make similar predictions have domains that map to similar areas in the space. Here, $D_{\mathcal{Y}}( \cdot || \cdot )$ can be any normalised distance metric over $\mathcal{Y}$ and is used to calculate the *Predictive Similarity Score* between two models evaluated on the same feature. This loss aims to minimise the distance $||f_\theta(\hat{x}_i) - f_\theta(\hat{x}_j)||^2$ - which is a proxy for the distance between the domains of models $i$ and $j$ - when the predictions between the two models are more similar. This is based on the intuition that if models are making randomly incorrect predictions for a feature they are unlikely to agree with other models (Dror et al., 2017). Thus, these points are more likely to be correctly classified - suggesting some overlap in training domain of the two models.

Given the non-negativity of distances we will need to balance the previous loss, which will always aim to reduce the distance between all the pairs in the loss, albeit up-weighting those that have more similar predictions between models. Thus, we employ $\mathcal{L}_{separation}$:

$$\mathcal{L}_{sep} = - \sum_{(\hat{x}_i, \hat{x}_j) \in \mathcal{D}_M \times \mathcal{D}_M} \frac{1}{2}\mathbb{1}\{\hat{x}_i(m) \neq \hat{x}_j(m)\} \times ||f_\theta(\hat{x}_i) - f_\theta(\hat{x}_j)||^2, \tag{4}$$

---

[2]It is possible to sample a larger model dataset to provide a lower variance estimate of the loss at each step, but given the pairwise distances we will calculate this can become computationally challenging.

which is designed so that models are naturally moved away from each other - essentially encoding a prior that data distributions for the different models will be distinct. With $\mathbb{1}$ denoting the indicator function and $x(m)$ denoting the model density from which the point was sampled, this loss pushes apart points that were not sampled from the same model. These losses can be balanced with weighting hyperparameters, the potential optimisation of which is discussed in the appendix.

### 3.3 Weights Estimation

Once a space is learnt, we can use it to make predictions. Given model densities in the feature space, denoted $p_j^{\mathcal{X}}(x)$, we construct a corresponding density in the representation space $p_j^{\mathcal{Z}}(z)$ - this can be achieved simply by sampling from $p_j^{\mathcal{X}}(x)$, passing through $f_\theta$ and modelling the new density with a kernel density estimate. From here, we calculate weights as the relative density a feature representation has under the densities in the new space:

$$w(x)_i = \frac{p_i^{\mathcal{Z}}(f_\theta(x)) + \gamma}{\sum_{j=1}^N p_j^{\mathcal{Z}}(f_\theta(x)) + \gamma}, \quad (5)$$

---

**Algorithm 1:** Synthetic Model Combination

**Result:** Test predictions using mapping from data to model weights

**Input:** $\{(\mathcal{M}_j, \mathcal{I}_j)\}_{j=1}^N$ and $\mathcal{D}_T$;

1. Use information to produce density models;
2. Sample data from models and combine with test data;
3. Learn representation space;
4. Re-model densities in new space;
5. Calculate weights in new space;
6. Make predictions $\{\hat{y}_i\}_{i=1}^M$ over test set;

**Return:** $\{\hat{y}_i\}_{i=1}^M$

---

with $\gamma$ a regularisation hyperparameter chosen to be very small such that an outlier's weights are not dominated by the closest model. The quantity $\sum_{j=1}^N p_j^{\mathcal{Z}}(f_\theta(x))$ can be used to inform the confidence of any prediction made by SMC. Particularly low values will indicate that the feature had low density under all the domains and as such it may be likely that none of the models were accurate. We note as well that assuming a hierarchical generative model for the test data where one of the models training data distributions is selected and then sampled from - this can be interpreted as the posterior probability that a test instance was sampled from a model's domain and is thus well represented by it.

## 4 Experimental Demonstration

In this section we will use a series of experiments to make the following concrete points about our method: **1)** In common scenarios, global ensembles *do not work*, and we must make instance-wise predictions (Section 4.1); **2)** Doing so in even slightly high dimensions *requires* a representation learning step, and our proposed losses improves the quality of the learnt representation (Section 4.2); **3)** We can make good predictions with *surprisingly little* information (Section 4.2); **4)** This is useful beyond synthetic example setups in *real-world case scenarios* (Section 4.3); **5)** Naturally, there are setups where we will *underperform*, which we should understand (Section 4.4); and **6)** We are *completely agnostic* to the type of models used (Section 4 - we demonstrate across a variety of models from simple regressions to convolutional nets and differential equations).

### 4.1 A Simple Regression Example

In this first example, we aim to demonstrate why this approach is a clear necessity when making future predictions, showing that we need to construct ensembles with an instance-wise approach to generate predictions that are accurate and appropriately calibrated. In our examples, we simulate feature data from experiment-dependant distributions and targets from a simple noisy sin wave curve.

**Instance-wise ensembles are necessary under our assumptions.** In our first example we consider two models: both simple neural networks, one trained on features sampled from a Gaussian centred at 5, with the other centred at 15, and standard deviation 3.5 - as can be seen from Figure 3**a** there is very little overlap in the support of the training data for the two models, and it can be seen that Model 1 clearly makes appropriate predictions when $x < 10$, while it is Model 2 that is accurate when $x > 10$. This relationship cannot be captured by a global ensemble of the two methods, as seen in Figure 3**b** - whereas SMC is perfectly able to do so.

**Our uncertainty can show when we are not confident.** In our previous example, SMC was able to match the target function across the feature space given the distribution of the training

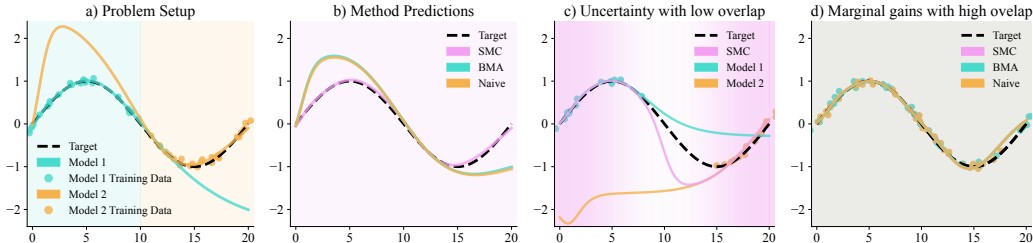

Figure 3: **Regression Example.** In all cases the background colour reflects the optimal model. This problem setup (**a**) demonstrates the key need of an instance-wise ensemble. It can be seen from the model predictions (**b**) that SMC is the only ensemble that can capture the target function. Altering the setup such that training domains are further away (**c**) shows our uncertainty (background colour concentration) can helpfully highlight where we should not be confident. Finally, when domains completely overlap (**d**) we can see all ensembles perform equally well and SMC loses its edge.

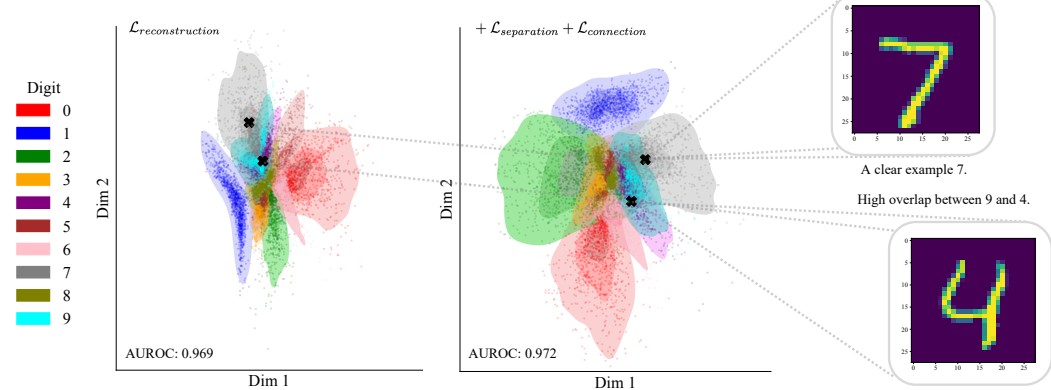

Figure 4: **MNIST Example.** Representation spaces learnt from MNIST using just the *reconstruction* loss as well as adding the *separation* and *connection* losses. Model information consisted of feature-only samples from the model training data. Test points are plotted to show their relationship to the model densities, showing that in low-dimensional space they usually have high density under a model.

features of the two models. What happens though if *none* of the models make accurate predictions on an instance because it is outside all of their domains? We explore this by moving the two Gaussians used to generate features for the two models further apart, centring them at 0 and 20 respectively - resulting in a region around 10 where none of the models make correct predictions, and consequently SMC's predictions also suffer as seen in Figure 3**c**. However, using the uncertainty explained in Section 3.3 (which is visualised by the background colour of the plot) we can see that the accuracy of SMC is calibrated well against its confidence. SMC can provide useful information about which parts of the feature space it is relatively less confident on.

## 4.2 Higher Dimensions with MNIST

In order to develop further points of our method, we move to a more complicated example with the most iconic ML problem of handwritten digit classification. Using MNIST ($d = 784$), we construct a problem where ten different classifiers are trained to each individually identify a single digit effectively while their performances on other digits are significantly lower - this is achieved by providing mostly only data of the respective single digit. The information provided per model involves a number of feature samples from the training set (number depending on exact experiment).

**High dimensions need to be reduced.** As discussed earlier, density estimation becomes ineffective in high dimensions. As the space gets bigger, the relative proportion covered by the data reduces. We find that even in the case where the full set of features (but of course not targets) used to train the predictive models are made available to SMC, *none* of the test set features have non-zero density to numerical precision - and hence a meaningful prediction cannot be made. On the other hand, once we incorporate a representation learning step, the coverage of the training examples significantly increases. This can be seen in Figure 4, which shows two separate 2-dimensional learnt

representation spaces: on the left using only the $\mathcal{L}_{rec}$ loss; and on the right including the $\mathcal{L}_{sep}$ and $\mathcal{L}_{con}$ losses. Here, test features are plotted as points, while the training features used per model are represented as density estimates. It can be very evidently seen that the majority of test features have significant density under at least one of the models in both of the example representation spaces learnt.

**Standard representation learning is good - but we are better.** We have established that some low-dimensional representation learning is necessary for problems like these - the fact is that by simply incorporating a standard representation learning step, using a standard autoencoder for example, will allow us to make reasonable predictions, indeed we can achieve a OneVsRest Area Under the Receiver Operating Characteristic curve (AUROC) of 0.969 on the test set. By including the extra regularisation though we can improve the ability of our algorithm. We can immediately see with a visual inspection of Figure 4 and the two learnt representation spaces that their inclusion results in more spread out and differentiated densities (both are plotted on the same scale). Additionally, we see and increase in the AUROC to 0.972. This appears to allow more appropriate evaluation of points like the example '4' in Figure 4, which could reasonably be mistaken for an incomplete '9'.

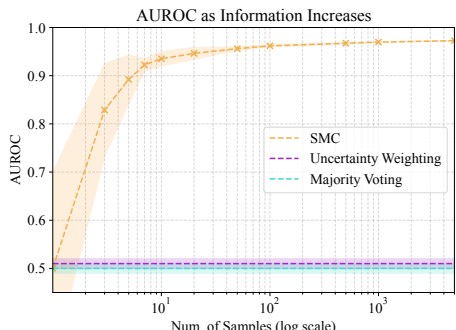

Figure 5: **Information Gain to Performance Relationship.** The AUROC obtained by SMC against the number of training data feature samples given to the method on a log scale.

**Many examples are unnecessary for significant performance gain.** In Figure 5 we plot the predictive AUROC of SMC against the number of feature samples from the training data passed to the method. In this setup the information provided to SMC are a number of feature examples per model that can be sub-sampled from to evaluate the regularisation losses and then used to construct density estimates in the representation space. Naturally, as the number of samples increases so does the quantity of information and so too does the performance of SMC. In the extremes, of course with no samples no meaningful prediction can be made, and when all the training features are provided an AUROC of about 0.97 is achieved. Interestingly, we can see that actually not a very large number of samples are required to obtain strong performance here. Notably, after only 3 samples are provided, SMC sees significant performance improvement to 0.83, compared to the baseline of 0.50. This is not a significant amount of information, with just three features and none of the target details.

**Further failure of global ensembles.** We include some additional global ensemble baselines in Figure 5 to demonstrate that they fail in more general settings than the synthetic example we previously discussed. We include a simple majority voting ensemble as well as a method that weights predictions based on a measure of the uncertainty of the ensemble member. In particular, we weight proportional to the exponential of the reciprocal of the entropy of the predicted categorical distribution.

### 4.3 Case Study: Vancomycin Precision Dosing

Population pharmacodynamic (PopPK) models are differential equations that model the concentration of a drug in the bloodstream of a patient. Vancomycin one of the most common antibiotics - with multiple PopPK models (Broeker et al., 2019) having been developed on different patient populations. For many drugs, including vancomycin, the area under the curve (AUC) of blood concentration over time is an important marker in estimating the effectiveness of a drug intervention, and can be used to support dose individualisation by predicting drug response in the future. An example of the models are found in Figure 6, showing the predicted concentration levels in an example patient for a number of different PopPK models.

**SMC can be applied to real challenges.** This pharmacological setting is one where SMC would be a

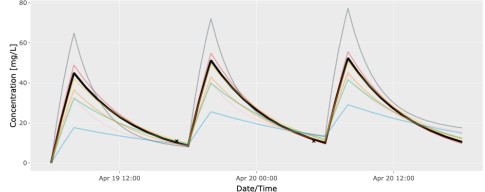

Figure 6: **Example Patient Blood Concentration.** Predicted concentration levels in the blood of an example patient. Each line represents a different model with the black being an ensemble. The area under the curve (AUC) is an important target to estimate the impact the drug will have, as it reflects the total exposure to the patient, and one that is impossible to calculate directly without continuous monitoring. Graphic produced using the TDMx (Wicha et al., 2015) software.

Table 2: **Model Performance** for predicting the AUC given varying numbers of blood concentration observations. We report relative root-mean-square error (RMSE) - note: *lower* scores are better.

| Model | A priori | One | Two | Three | Four |
|---|---|---|---|---|---|
| Adane et al. (2015) | $80.0 \pm 0.7\%$ | $47.3 \pm 0.4\%$ | $40.9 \pm 0.6\%$ | $32.5 \pm 0.3\%$ | $29.4 \pm 0/3\%$ |
| Mangin et al. (2014) | $83.3 \pm 0.6\%$ | $36.2 \pm 0.3\%$ | $33.0 \pm 0.3\%$ | $21.4 \pm 0.2\%$ | $15.6 \pm 0.2\%$ |
| Medellín-Garibay et al. (2016) | $76.5 \pm 0.6\%$ | $33.5 \pm 0.2\%$ | $28.5 \pm 0.2\%$ | $20.9 \pm 0.2\%$ | $17.4 \pm 0.2\%$ |
| Revilla et al. (2010) | $\mathbf{32.7 \pm 0.4}\%$ | $21.6 \pm 0.2\%$ | $20.0 \pm 0.2\%$ | $16.7 \pm 0.2\%$ | $15.3 \pm 0.2\%$ |
| Roberts et al. (2011) | $35.8 \pm 0.3\%$ | $\mathbf{20.6 \pm 0.1}\%$ | $20.0 \pm 0.2\%$ | $16.1 \pm 0.1\%$ | $14.5 \pm 0.2\%$ |
| Thomson et al. (2009) | $48.2 \pm 0.4\%$ | $30.2 \pm 0.2\%$ | $25.8 \pm 0.2\%$ | $20.9 \pm 0.1\%$ | $19.1 \pm 0.2\%$ |
| Model Averaging | $55.7 \pm 0.4\%$ | $28.5 \pm 0.2\%$ | $24.9 \pm 0.2\%$ | $19.3 \pm 0.2\%$ | $16.6 \pm 0.2\%$ |
| **SMC** | $41.8 \pm 0.5\%$ | $22.1 \pm 0.2\%$ | $\mathbf{19.6 \pm 0.2}\%$ | $\mathbf{15.2 \pm 0.2}\%$ | $\mathbf{12.8 \pm 0.1}\%$ |

particularly appropriate method of choice. First, the private medical nature of the problem means that data on drug response in humans is not widely available, and so new models cannot be trained on the data of the previous models. Second, the datasets that the models are built on are often relatively small and focus on a specific subpopulation, such as those at a particular hospital, or suffering a specific comorbidity. Third, researchers usually publish the model alongside demographic information on the patients used to produce them, fulfilling SMC's need for both a model and information.

**Accurate Drug Response Estimation.** We base our experiment around those of Uster et al. (2021) who themselves consider an ensembling approach through the application of model averaging. We use simulated patients provided by the authors to evaluate the effectiveness of SMC in the accuracy of predicting the AUC across a number of settings when a number $\in \{0 \text{ (A priori)}, 1, 2, 3, 4\}$ of concentration measurements are taken in a 36-hour period. Ultimately, we have six models, each from a separate subpopulation {extremely obese, critically ill post heart surgery, trauma patients, intensive care patients, septic, hospitalised patients}, as well as a variety of demographic information for each[3]. In our experiments we focus on the age, height, weight, sex, and creatinine clearance levels as have been shown to be strongly associated with drug response (Uster et al., 2021) and are provided for each model. In Table 2 we report the relative root-mean-square error RMSE of the predictions - the lower, the better. We can see that SMC consistently performs well - and indeed performs the best when at least two observations are made available to the methods.

## 4.4 Understanding Challenging Scenarios for SMC

We must accept that there's no such thing as a free lunch, and SMC does not have the answer for everything - there are scenarios where it may underperform for some reason other than obvious cases when model information is not available or where individual models are themselves bad.

**Performance discrepancy between models leads to worse predictions.** Table 2 highlights a situation where SMC may underperform. We see that SMC performs *worse* when there is *high variability* in the performance of individual models. For example, in the *A priori* setting, there is a very large range in RMSE, from 32.7 all the way up to 83.3. Since SMC does not attempt to evaluate the relative performances of the models, when there are models that just perform very badly they can severely detract from SMC's performance. This highlights that SMC performs best when all the models perform well *in their respective domain*, but that those domains are relatively *disjoint*. Could you do anything if you know (through perhaps a validation set) that some models perform very badly? A simple solution would involve also calculating the weights given by BMA as well, before averaging both sets of weights. This would weight models globally by some level of how confident we are that the model is good as well as locally by how well we believe the model will be able to perform on a specific feature, balancing the potential causes of poor performance.

**Domain overlaps result in marginal improvement.** To examine this point we shall revisit the regression example of earlier. SMC assumes that the domains of the different models are at least partially different so that the feature dependant weights allow for selecting the model(s) that are most appropriate. This benefit is lost when all the model domains are the same, as we show in Figure 3**d** - all the models are equally good across the feature space and so an ensemble will perform just as well as SMC. It is important to note SMC does *not* perform worse, it just does not perform better.

---

[3]Further information on the setup, including models and associated information can be found in the appendix.

# 5 Discussion

**Society and Ethics.** The setting of our method is primarily focused on situations where the sharing of data is in some way tricky or limited - a key example being the medical regime. As such, our method is designed to be able to perform well in this area and so would hopefully have an overwhelmingly positive impact. Of course as with any method there is potential for it to be misused in an application which has damaging effects, but it seems unlikely that SMC poses any intrinsic danger.

**Conclusions.** In this paper we have introduced the framework of Synthetic Model Combination - an instance-wise approach to unsupervised ensemble learning, having established that there are many cases when global ensembles are simply inadequate for meaningful predictions. We additionally introduced a novel unsupervised representation learning method for the sparse high-dimensional setting, and showed the use of our method in both example synthetic problems and the real case study of estimating the effectiveness of vancomycin precision dosing.

## Acknowledgements

AJC would like to acknowledge and thank Microsoft Research for its support through its PhD Scholarship Program with the EPSRC. This work was additionally supported by the Office of Naval Research (ONR) and the NSF (Grant number: 1722516). Thanks to Sebastian Wicha and David Uster for helpfully providing simulations from their own research for the vancomycin example, as well as Jean-Baptiste Woillard for encouraging us to examine this problem. We would also like to thank all of the anonymous reviewers on OpenReview, alongside the many members of the van der Schaar lab, for their input, comments, and suggestions at various stages that have ultimately improved the manuscript.

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
