Table 3: **Vancomycin population demographics** for each of the models that we consider. This information was published in the original papers where the models were also presented.

| Model | Age | BMI | Body Weight | Sex | CrCl |
|---|---|---|---|---|---|
| Adane et al. (2015) | $43.0 \pm 7.5$ | $49.5 \pm 5.2$ | $147.9 \pm 13.1$ | 0.61 | $124.8 \pm 14.0$ |
| Mangin et al. (2014) | $63 \pm 23$ | $28 \pm 7$ | $82 \pm 21$ | 0.87 | $138 \pm 50$ |
| Medellín-Garibay et al. (2016) | $74.3 \pm 14$ | $27.5 \pm 5$ | $72 \pm 15$ | 0.45 | $90.5 \pm 52$ |
| Revilla et al. (2010) | $61.1 \pm 16.3$ | $26.2 \pm 4.1$ | $73.0 \pm 13.3$ | 0.66 | $86.1 \pm 55.1$ |
| Roberts et al. (2011) | $58.1 \pm 14.8$ | $25.9 \pm 5.4$ | $74.8 \pm 15.8$ | 0.62 | $90.7 \pm 60.4$ |
| Thomson et al. (2009) | $66 \pm 20$ | - | $72 \pm 30$ | 0.63 | $98 \pm 51.0$ |

# Appendix

# A   Further Experimental Details

All experiments were performed on a 2021 MacBook Pro, using an Apple M1 Pro chip with 16 GB of RAM.

Code was written in PyTorch (Paszke et al., 2019), and hyperparameters for regular (non SMC) methods were selected through grid search over a validation fold of the training data where appropriate.

## A.1   MNIST

As predictive models we trained 10 neural networks with a unified architecture consisting of: two convolutional layers followed by two fully connected layers with ReLU activations. Each model was trained on a tenth of the full training set consisting of a 90/10 random split of one particular digit vs a random selection of training examples.

The SMC representation network consisted of a symmetric encoder/decoder architecture of three fully connected layers with ReLU activations between them.

## A.2   Vancomycin Precision Dosing

In table 3 we show the population demographics information that was used for each of the published models used in our experiments. In Thomson et al. (2009), there was no BMI information, so we took the average across all the other models for the basis of our calculations. With this information, we used factorised Gaussian distributions for the continuous covariates and Bernoulli distributions for the binary ones to construct density estimates.

The simulations used for evaluating the methods were provided to us by Uster et al. (2021), who generated patients solving the differential equations using NONMEM (Beal and Sheiner, 1979).

However, the simulation setup they use is not based on the underlying assumption that we make. I.e., when simulating patients based on the model of Adane et al. (2015) for clinically obese patients, the current simulations still generate covariates from a normal population, and actually only a small minority of the patients would be considered obese. Consequently, in order to evaluate the performance of SMC in what we consider a more realistic setting we develop a method to subsample the original simulations in order to obtain a population for each model that more accurately reflects the population on which each model was developed.

In order to select a smaller sample of 1000 patients, we first modelled the density of each of the patient populations based on the demographic statistics provided in each of the original papers. Then for each of the 6000 simulated patients we evaluated the likelihood that their covariates came from each model and selected the model with the highest likelihood. If this selected model matched the model from which the AUC observations were simulated, then the patient was kept and otherwise discarded. This mimics a rejection sampling method for the covariates from the original model demographics using the sampling method of Uster et al. (2021) as the base distribution. This results in a population where each model only simulated data for patients whose covariates were likely under their reported demographic information.

# B    A Note on Hyperparameter Optimisation

In general, the problem of hyperparameter optimisation in the setup that we consider is just as tricky as the main problem of determining which the best models are - without validation data it is hard to know which hyperparameters are necessarily better than the others as clearly the loss values on the training data alone do not indicate the ability of any model to generalise.

When searching for network hyperparameters for the SMC networks, it is possible to use standard methods while holding out part of $\mathcal{D}_T$, as it is possible to optimise the reconstruction loss to see which networks are effective at learning some form of useful representation.

Balancing the losses is more challenging since the main aim is to optimise the final prediction accuracy, and without some held out validation data it is not possible to get a good idea of this. We found experimentally that an equal weighting across all the losses worked well, but that we could improve performance by increasing the weighting of the $\mathcal{L}_{con}$ and $\mathcal{L}_{sep}$ losses while keeping the optimised $\mathcal{L}_{rep}$ loss within some threshold (typically about 2.5%) of its value when the other losses are not included.

# C    A Note on Computational Complexity

For the training time, we essentially inherit the properties of a variational autoencoder, although the calculation of the regularisation terms requires calculating pairwise-distances between points, an operation that is $\mathcal{O}(n \log n)$ with $n$ the mini-batch size, something which should be taken into account. This is very manageable, especially as there is no need for this to be repeated or updated and the applications we see this being useful for are not time pressured.

Inference is fast, taking only a single pass through a network followed by calls to low dimensional densities. The initial pass scales only linearly with input dimension, and the calls to the densities depend on the exact method of estimation, but realistically will be fast given their lower dimensionality. Again though, we consider most applications for this to not require a particularly fast inference time anyway.

# D    Future Directions

We hope that this paper encourages further work in the area of unsupervised ensemble learning, given its relatively underexplored state. We take a very general approach in this work, and it is very likely we could improve performance given a larger restriction on the domain information we use, e.g. always receiving a kernel density estimate.

One of the largest problems in imitation learning is the drift in states caused by compounding small errors in a policy (Ross and Bagnell, 2010) which leads an agent to areas of a state space where they are not familiar and hence are unlikely to act optimally. One solution is interpretable policy methods (Pace et al., 2021; Chan et al., 2022) so that we can inspect the policy and hopefully notice if it is about to take actions which are not sensible. Alternatively, a Bayesian approach in order to account for uncertainty (Chan and van der Schaar, 2020) can be helpful to identify parts of the statespace where we are unsure about their value.

These methods rely on some level of experience in the environment (mostly from expert demonstrations), and the challenge is clearly harder if given just suggestions from an expert policy. Deploying multiple expert policies with information about the areas in which they are confident based on SMC could offer a solution, although it presents more challenges as we need to describe areas of expertise over *trajectories* and not just single time steps.