# OpenReview forum: "Synthetic Model Combination: An Instance-wise Approach to Unsupervised Ensemble Learning"
_NeurIPS.cc/2022/Conference — NeurIPS 2022 Accept_

### Official Review · Reviewer_6vwo · 2022-07-10

**Rating:** 5
**Confidence:** 3
**Soundness:** 2 fair
**Presentation:** 3 good
**Contribution:** 3 good

**Summary:**

This paper proposed a new ensemble method – Synthetic Model Combination (SMC) operating on the instance-wise level in the setting of unsupervised ensemble learning. The authors also introduced a novel unsupervised representation learning method. The experimental results on MNIST and Vancomycin Precision Dosing datasets validate the effectiveness of the proposed methods.

**Questions:**

1. Some figures and tables (e.g., Figure 1, Table 1, Algorithm 1) are not mentioned in the main body. The authors are suggested to link all the figures and tables to the main body.
2. The authors are suggested to make more clarifications for some descriptions or functions. For example, you introduced p_{j}^{\chi}(x) in line 212. More details about it are suggested to be added.


**Limitations:**

Yes.

**Strengths And Weaknesses:**

Strengths:
1. This paper is well-written.
2. The problem this paper focuses on is important, and the method is interesting.


Weaknesses:
1. In this paper, the solid theoretical analysis (e.g., theorems) is missing. I think it is a necessary part, especially for a top machine learning conference like NeurIPS.
2. Instance-wise level methods are expected to perform better than the methods learning the fixed weights for every instance, but at the cost of higher time complexity. The part about the time complexity analysis is suggested to be added.

---

> ### Author Response · Authors · 2022-08-02
> **Author Response**
>
> Thank you very much for your helpful feedback and comments. We aim to address all of the individual points in your review here but please also see the revised manuscript for changes.
>
> *”In this paper, the solid theoretical analysis (e.g., theorems) is missing…”*
>
> **Response**:
>
> We would like to respectfully disagree with the premise that the manuscript should contain more theorems in order to be appropriate for a venue like NeurIPS. A very large proportion of the accepted papers at these top conferences do not contain such aspects as they focus on algorithmic advances, applications, or empirical studies - all of which advance our knowledge of the machine learning field effectively.
>
> We are also not sure what theorems there are in our case that would provide some significant insight into the method.  We aim to focus on a new problem that has been underappreciated by the research community thus far, and have proposed a novel solution to this. We have demonstrated experimentally that this method works well in the settings we are interested in, settings that existing methods fail significantly, and that they can be applied easily to a wide range of settings. We believe this would be of great use to the machine learning community in and of itself and that NeurIPS would be an excellent venue where many people might be interested in the work.
>
> If you have any suggestions on what theory we could add in order to better understand the method or draw particular insight that we have not been able to demonstrate with our experiments then please do let us know as we are keen to build the understanding surrounding this method.
>
> *”The part about the time complexity analysis is suggested to be added.”*
>
> **Response**:
>
> Thank you for the suggestion on including details on computational complexity. We have added a section in the appendix of the paper for the moment, but with the additional page provided in the camera ready version we can add it to the main paper. We also give a summary here:
>
> For the training time, we essentially inherit the properties of a variational autoencoder, although the calculation of the regularisation terms requires calculating pairwise-distances between points, an operation that is $\mathcal{O}(n\log{}n)$ with $n$ the mini-batch size, something which should be taken into account. This is very manageable, especially as there is no need for this to be repeated or updated and the applications we see this being useful for are not time pressured.
>
> Inference is fast, taking only a single pass through a network followed by calls to
> low dimensional densities. The initial pass scales only linearly with input dimension, and the calls to the densities depend on the exact method of estimation, but realistically will be fast given their lower dimensionality. Again though, we consider most applications for this to not require a particularly fast inference time anyway.
>
>
> *”Some figures and tables (e.g., Figure 1, Table 1, Algorithm 1) are not mentioned in the main body…”*
>
> **Response**:
>
> Thank you for pointing this out, we have made sure that references to all figures/tables/algorithms now appear in the main text, and not just footnotes as some were.
>
> *”The authors are suggested to make more clarifications for some descriptions or functions…”*
>
> **Response**:
>
> Thank you for pointing this out, we will clarify these points in the paper, especially on line 212 where we now make the point that this is how we shall denote this object. We have made a small adjustment now - but with the additional space in the camera ready version we can really take the space to emphasise this.

---

> ### Author Response · Authors · 2022-08-05
> **Follow Up**
>
> Dear Reviewer 6vwo,
>
> Once again, thank you for your thoughtful review. While we have this current period of discussion please do let us know if our response has addressed your concerns - we are keen to keep engaging with you to address any additional questions or comments.
>
> Best wishes,
>
> The Authors

---

### Official Review · Reviewer_gNf4 · 2022-07-10

**Rating:** 4
**Confidence:** 3
**Soundness:** 2 fair
**Presentation:** 2 fair
**Contribution:** 2 fair

**Summary:**

This paper proposed an approach named Synthetic Model Combination (SMC) that make weighted ensemble predictions based on local manifold context. In addition, the authors proposed a representation learning procedure to obtain the specified weights for SMC. The author provided examples and benchmarks on synthetic and real world datasets at the end.

**Questions:**

-It would be interesting to see how encoding simple uncertainty measures like variance or entropy as weights compares with weights calculated from VAE. Have the author performed similar experiments as a control?

-Could the author clarity "augmentation of the individual models" at line 46?

-What is the added computation complexity of SMC?

**Strengths And Weaknesses:**

Strengths:

-This work raises an important question that the ensemble process should be localized due to the partial information received during the training process of the contributing models.

-Description of the problem is mostly clear and the article is well structured.

Weaknesses:

-Not novel.
The main contribution proposed in this article is a weighted ensemble on individual samples based on local context. This idea has been explored in previous studies (Huang 2009, Weighted Average with Data Dependent weights). The proposed autoencoder is new but seems to only be a way of obtaining weights. Authors need to state the added advantages of using an VAE. If it is the best way to obtain weights the author need to compare autoencoder with other methods to justify this approach.

-Results not significant/lack of benchmarks:
The only benchmark is on a drug response estimation with incremental improvements. To demonstrate a general performance improvements, experiments on existing larger dataset like cifar10, imagenet, and other NLP datasets is needed.

-Lack of comparison with existing methods:
Similar to the previous point. The author should systematically compare the SMC with existing ensemble learning approaches across multiple benchmarks.

-The overall writing quality is not great. There are many non-idiomatic sentences with long clauses that hinders readability.

-While I appreciate the illustration with well-colored diagrams, it can be a little distracting at times. Figure components should be explained in the figure legend. (e.g. dots in Figure 3.)

Ref:
Huang, Faliang, Guoqing Xie, and Ruliang Xiao. "Research on ensemble learning." *2009 International Conference on Artificial Intelligence and Computational Intelligence*
. Vol. 3. IEEE, 2009.

---

> ### Author Response · Authors · 2022-08-02
> **Author Response 1 of 2**
>
> Thank you very much for your helpful feedback and comments. We aim to address all of the individual points in your review here but please also see the revised manuscript for changes.
>
> *Novelty*
>
> **Response**:
>
> We would like to strongly contend the suggestion that our work is not novel.
>
> In Huang et al. (2009), they point to the work of Verikas et al. (1999) which describes a method for weighted averaging with data dependent weights.
>
> Their method works by subdividing the space into $k$ regions, and then finding the optimal weights *in that region* - thus some different examples will have different weights used but we can’t consider it truly instance-wise, only region-wise.
>
> More importantly however, this method is *not* unsupervised - in order to calculate the weights in each region they state very clearly on page 435 that “we use a validation set to evaluate the performance” - and so this method cannot work in the settings that we describe, and so it cannot be that our work is doing the same thing.
>
> That being said, we appreciate you pointing out this work and will add a discussion of it to the paper as the inclusion of data-dependent weights outside of the unsupervised setting should provide some useful background.
>
> *References*:
>
> Huang, F., Guoqing X., and Ruliang X., (2009). Research on ensemble learning. International Conference on Artificial Intelligence and Computational Intelligence . Vol. 3. IEEE.
>
> Verikas, A., Lipnickas, A., Malmqvist, K., Bacauskiene, M., & Gelzinis, A., (1999). Soft combination of neural classifiers: A comparative study. Pattern recognition letters, 20(4), 429-444.
>
> *On benchmarks and baselines*
>
> **Response**:
>
> We have tried hard to design all of the experiments in the penultimate section of the paper to answer specific questions about the method and to allow a reader to better understand both the strengths and weaknesses of the method - presenting results not just on drug response but also MNIST and regression problems. Given the limited space we thought it better to focus on this rather than a repeated demonstration that SMC is the only method that works in the settings described, albeit using different base datasets.
>
> It ultimately comes down to the fact that we do not know of any other unsupervised methods that make instance-wise predictions and thus we don’t believe there are any methods for which it makes sense to compare performance across a wide range of benchmarks. Simply put, if methods are not instance-wise then they will perform poorly on the benchmarks we use (which will be adaptations of existing datasets where the training domains of the individual models will be disjoint).
>
> We hope you find this position compelling, although please let us know if you still have concerns. In the meantime however we have also added some more baselines to the MNIST experiments - including as we describe next, your suggestion for an uncertainty based weighting.
>
> *Uncertainty based weighting*
>
> **Response**:
>
> Thank you for this suggestion, as we think it will be very helpful to demonstrate some of the finer points of our model. We have now included results for this in the MNIST example that can now be seen in Fig 5. As you can see, it performs quite poorly - this should not be particularly surprising though since neural networks are notoriously bad at calibrating their uncertainty in the first place (Guo et al. 2017). We should also note that they are specifically capturing *aleatoric* uncertainty (i.e. the inherent noise in the data), which unlike *epistemic* uncertainty will not tell us about whether the network has seen anything like the example in its training data and thus whether it may or may not be a useful model for making a prediction over this specific example.
>
> Supposing the models given to us were Bayesian models, we should be able to get further with this approach, but this is obviously significantly limiting, as we expect it less likely for groups to publish full posteriors over parameters, and is not something we have seen happen in the real world to a large extent. It’s worth noting as well that at scale, variational methods for Bayesian inference over neural network weights have been shown to result in poorly calibrated posterior predictive distributions as well given the approximations made in the posterior, so even there we may expect over-confidence (Ovardia et al. 2019).
>
> *References*:
>
> Guo, C., Pleiss, G., Sun, Y. and Weinberger, K.Q., (2017). On calibration of modern neural networks. International conference on machine learning (pp. 1321-1330). PMLR.
>
> Ovadia, Y., Fertig, E., Ren, J., Nado, Z., Sculley, D., Nowozin, S., Dillon, J., Lakshminarayanan, B. and Snoek, J., (2019). Can you trust your model's uncertainty? evaluating predictive uncertainty under dataset shift. Advances in neural information processing systems, 32.

---

> > ### Author Response · Authors · 2022-08-02
> > **Author Response 2 of 2**
> >
> > *”Could the author clarity "augmentation of the individual models" at line 46?”*
> >
> > **Response**:
> >
> > For this, we simply mean that we do not alter the given models in any way - for example, we wouldn’t take models as pre-trained starting points and then refine them on more data.
> >
> > *”What is the added computation complexity of SMC?”*
> >
> > **Response**:
> >
> > In terms of details on computational complexity, we have now added a section in the appendix of the paper for the moment, but with the additional page provided in the camera ready version we can add it to the main paper. We also give a summary here:
> >
> > For the training time, we essentially inherit the properties of a variational autoencoder, although the calculation of the regularisation terms requires calculating pairwise-distances between points, an operation that is $\mathcal{O}(n\log{}n)$ with $n$ the mini-batch size, something which should be taken into account. This is very manageable, especially as there is no need for this to be repeated or updated and the applications we see this being useful for are not time pressured.
> >
> > Inference is fast, taking only a single pass through a network followed by calls to
> > low dimensional densities. The initial pass scales only linearly with input dimension, and the calls to the densities depend on the exact method of estimation, but realistically will be fast given their lower dimensionality. Again though, we consider most applications for this to not require a particularly fast inference time anyway.
> >
> > *"Figure components should be explained in the figure legend."*
> >
> > **Response**:
> >
> > Thank you for pointing this out, we have now updated the legends to reflect this

---

> > ### Comment · Reviewer_gNf4 · 2022-08-07
> > **Comment to Response 1**
> >
> > I thank the authors for their effort in editing to the manuscript.
> >
> > 1. Novelty
> > I agree that unsupervised + instance-wise weighting is a new combination. However a more thorough literature review on instance-wise weighting is needed. Again, I am sure each new method will differ from existing methods to some degree, but the author should properly refer to previous studies so that readers can better position the study and the problem being investigated.
> >
> > 2. Uncertainty based weighting
> > Could the author explain why increase sample size does not increase uncertainty performance in Figure 5? I found this very counter intuitive.

---

> > > ### Author Response · Authors · 2022-08-07
> > > **Response**
> > >
> > >
> > > Thanks a lot for the response, we really appreciate your engagement with our work as we know this can be a very time consuming task!
> > >
> > > *Novelty*: We are very glad you agree! And we also agree with you that a larger part of the related work should include this area of research - we have currently updated the manuscript to include this in order to highlight our intentions given current space constraints, but with the extra camera-ready page we can fully explore this. If you have any additional suggestions on work to include it would be gratefully received!
> > >
> > > *Uncertainty*: Our apologies for the confusion - the added feature examples on that axis are the ones  given to SMC in order to estimate the densities, **not** the ones given to the ensemble models when training in the first place, and so is not a relevant axis for the uncertainty estimation benchmark and does not effect performance. The only reason we included them in Fig 5 and not separately was due to, as above, space constraints - we can, and intend to, expand and introduce this properly in the camera-ready version.
> > >
> > > If this has alleviated your concerns we kindly request you consider raising your recommendation, but if you have any further concerns, please do let us know!
> > >
> > > All the best,
> > >
> > > The Authors

---

> ### Author Response · Authors · 2022-08-05
> **Follow Up**
>
> Dear Reviewer gNf4,
>
> Once again, thank you for your thoughtful review. While we have this current period of discussion please do let us know if our response has addressed your concerns - we are keen to  keep engaging with you to address any additional questions or comments.
>
> Best wishes,
>
> The Authors

---

### Official Review · Reviewer_oUUV · 2022-07-11

**Rating:** 6
**Confidence:** 3
**Soundness:** 3 good
**Presentation:** 4 excellent
**Contribution:** 3 good

**Summary:**

This paper studies the problem of unsupervised ensemble learning in the problem setting where a set of trained models and limited unlabelled test instances are given, but no access to the data used for training the models is available. The aim of the ensemble learning is to obtain a set of optimal weights of the models specific to each given test/new instance, to form an ensemble for predicting the label of the instance.

A few demonstrating experiments, including one which is based on a real world medical problem, are presented to show the advantages of the proposed method and its limitations.


**Questions:**

1. What is the minimal requirement on the number of test samples?
2. What is the number of features of the dataset used in the “real case study” in Section 4.3?
3. Could you elaborate on D_Y in (3) a bit more, e.g. its role, calculation and justification on the multiplication used?
4. In terms of related work, I’d like to confirm - do all existing instance-wise or dynamic ensemble learning methods require model training data? Are there any methods which work in the same or similar setting of the proposed method? If yes, why the evaluation does not include a comparison study involving those methods?



**Limitations:**

Yes.

**Strengths And Weaknesses:**

Strength:
1. The paper addresses a practical problem in ensemble learning.
2. The research question is very well motivated and formulated.
3. The overall design of the method is reasonable and sound.
4. The experiment illustration has provided some strong evidence of the effectiveness of the proposed method.
5. The paper is well written.

Weakness:
1. The experimental demonstration, although carefully designed, with each example intended for a justifiable purpose, overall the evaluation is not strong. In particular, I have some concerns about “real case study”:

a. the data used for the “real case study” is simulated patient data, not real patient data, which makes people wonder whether the proposed method would perform effectively in real world cases or not.

b. the distributions of the predicted concentration by different models seem to have similar shapes/coverage, so I wonder if this is a good case study for demonstrating the expected advantage of the proposed method.

c. this case does not seem to be a high-dimensional case, hence the evidence provided by this case does not demonstrate the expected strength of the proposed method for dealing with high-dimensional data.

2. It is not clear as a minimal, how many unlabelled test instances are needed for learning the low-dimensional representations. For the experiment datasets, I cannot find information on the size of the test dataset either. A high requirement on test instances needed may limit the practical use of the proposed method.

Minor: In formula (5), w_i(x) should be w(x)_i, to be consistent with the notation used in Section 3 (outline of the three main steps)

---

> ### Author Response · Authors · 2022-08-02
> **Author Response**
>
> Thank you very much for your helpful feedback and comments. We aim to address all of the individual points in your review here but please also see the revised manuscript for changes.
>
> *Real case study details and "What is the number of features of the dataset used in the “real case study” in Section 4.3?"*
>
> **Response**:
>
> With respect to your questions on the real case study, yes the testing set was simulated as mentioned, but we should emphasise that the problem was a very real example that is currently being attempted to be solved in the pharmacological community. The point being here that the whole unsupervised ensemble learning is in fact a valid practical agenda and not just something we have fun with in made up settings.
>
> The setting was five dimensional, as we described in Appendix A - which we agree is not particularly high dimensional and so yes does not fully allow us to explore the advantages of the method in that respect. However we believe that we managed to demonstrate that aspect well in the MNIST example, with the Vancomycin example now reflecting a real problem and that there are gains to be had even without the dimensionality reduction.
>
> *”What is the minimal requirement on the number of test samples?"*
>
> **Response**:
>
> We do not think it is quite as simple as to just need a minimum number of test examples, as we are also expecting some information from the ensemble models’ training domains and so if that is in the form of feature examples they can also be used effectively to reduce the number of test examples needed. Actually describing what the minimum number is is still an open question, and with fewer examples it will be harder for the model to learn a good representation and could be prone to overfitting quite easily. That being said, with the added regularisation and the intention to use very low dimensional latent spaces, this should alleviate some of the issues, and combined with reducing the capacity of the encoder/decoder networks should make it practical in a reasonably wide range of scenarios.
>
> In terms of the test sets used in the paper, for MNIST we used the standard test set of 10,000 images, while the Vancomycin examples were validated on 6000 patients.
>
> *”Could you elaborate on D_Y…"*
>
> **Response**:
>
> D_Y is an important part of the regularisation and plays a large role in our representation step as it is what allows the representation to reason about models based on their agreement on inputs. It is essentially simply a measure over the disagreement between the different predictive distributions between the ensemble methods - as such its calculation will vary between tasks but in the case of classification we can use a simple normalised KL divergence between the predictive categorical distributions for example. The multiplication is simply there as this is used as a weighting over the pairs, up-weighting ones that have similar predictions.
>
> *”In terms of related work…"*
>
> **Response**:
>
> To the best of our knowledge the existing instance-wise and dynamic ensembles require training data in order to work, making them inappropriate for the settings we are interested in.
>
> Based on reviewer feedback however, we think it appropriate now to include a number of alternative methods in the MNIST example for clarity - even though, as can be seen in Fig 5 now, they perform very badly in the environment since they are not designed for this setting.
>
> *Minor notation*
>
> **Response**:
>
> Thank you for pointing this out, we have amended the notation in the manuscript.

---

> > ### Comment · Reviewer_oUUV · 2022-08-09
> > **Thanks for your response**
> >
> > Thanks the authors for your detailed response. The authors' explanation t on the "real case" makes good sense to me, but from the perspective of evaluating and demonstrating the proposed method, the "real case" may not provide strong evidence/confidence. Overall I feel that the method is promising and meanwhile there is room to improve the paper.

---

### Official Review · Reviewer_NtsN · 2022-07-11

**Rating:** 7
**Confidence:** 3
**Soundness:** 4 excellent
**Presentation:** 4 excellent
**Contribution:** 3 good

**Summary:**

The paper proposes synthetic model combination, an unsupervised ensemble learning approach to make instance-wise predictions by assuming access to the feature distributions the available models have perceived. Instance-wise, here, refers to choosing the weights of individual models for each test instance. The central application for such an approach is trained models on private (e.g. highly sensitive) data, where model performance is sufficiently good on the original data. The authors propose to learn a latent space on available unlabeled test data (sampled from the original data distributions), which in addition to a reconstruction loss also takes into account agreements and disagreements of the available models. The approach is evaluated for a toy example to underpin the need for the approach, MNIST and Vancomycin Precision Dosing. The results are able to show that instance-wise predictions using the learned latent space are performing well if the assumptions are met.


**Questions:**

* Which baseline methods have been tested for the MNIST use case?

* Did you test the approach for more challenging image / natural language datasets?

**Limitations:**

The paper discusses limitations of the approach, such as varying model performances. The approach therefore works best if there are individual expert models for their respective domain. If a model varies in performance in its domain and cannot dominate other models, there might be problems.

As mentioned before, it remains unclear if the dimensionality reduction step also works sufficiently well for more complex problems.

**Strengths And Weaknesses:**

Strengths: The paper is well-writen and thoroghly introduces/motivates the topic and problem setting. The problem is further concisely distinguished from related fields, such as transfer learning. The taken approach is well motivated too. The empirical evaluation is rather thorough and the results are sufficiently discussed, such that the available claims can be supported. While the general problem of unsupervised ensemble learning is known, the approach is sufficiently novel and original.

Weaknesses: A minor weakness wrt related work is missing coverage of existing unsupervised learning approaches which already incorporate agreement and disagreement based ensemble construction. Examples are [1,2]. It would also be interesting to see the performance of more challenging image or text datasets, but the current evaluation already supports central claims.

[1] Platanios, E.A., Dubey, A. and Mitchell, T., 2016, June. Estimating accuracy from unlabeled data: A bayesian approach. In International Conference on Machine Learning (pp. 1416-1425). PMLR.
[2] Platanios, E.A., Blum, A. and Mitchell, T.M., 2014, July. Estimating Accuracy from Unlabeled Data. In UAI (Vol. 14, p. 10).

---

> ### Author Response · Authors · 2022-08-02
> **Author Response**
>
> Thank you very much for your helpful feedback and comments. We aim to address all of the individual points in your review here but please also see the revised manuscript for changes.
>
> *"A minor weakness wrt related work..."*
>
> **Response**:
>
> Thank you very much for the suggested additional related work - we discuss the topic of unsupervised learning based on ensemble member agreement briefly in the manuscript (lines 99-102) but we will now include these additional references as they are certainly very relevant and bring more depth to the discussion.
>
>
> *"Which baseline methods have been tested for the MNIST use case?"*
>
> **Response**:
>
> The MNIST example is specifically designed to highlight that there are scenarios where global ensembles will fail significantly. Because of this we did not originally include the scores of other baseline methods as they didn’t seem to be a useful comparison, but based on reviewer feedback we have decided to include results on a few of the baselines, including majority voting and an uncertainty weighted ensemble. As we see though, they do quite badly since they produce global ensembles that are not appropriate for the task or have poorly calibrated and inappropriate uncertainty estimates.
>
> *"Did you test the approach for more challenging image / natural language datasets?"*
>
> **Response**:
> We haven’t completed extensive testing on larger image and natural language datasets, although have confirmed that SMC does work in these settings.
>
> The problem being that benchmarks for our task do not really exist - and so we would be simply repeating what we do with MNIST and taking an existing benchmark and processing it in a way that moved it into our setting, allowing us to control the level of shifts etc. in the data that each model in the ensemble was trained on, and engineering an example that SMC does well on. Thus, as long as autoencoders and neural classifiers can be applied to a benchmark, SMC should be able to output reasonable performance - which brings us to the second issue that we don’t believe that there are any appropriate instance-wise unsupervised emsembling methods to truly compare against.
>
> As such, we didn’t think that simply including scores of only SMC on a variety of datasets would be the most useful thing to include in the experiments section or actually give readers much information about how well our method works, and rather use the limited space to explore the properties of the method.

---

### Author Response · Authors · 2022-08-02
**Rebuttal Revision**

**Rebuttal Revision**

Dear Reviewers, thank you all for your thoughtful reviews and comments. We very much appreciate the general consensus that the problem we are working on and our proposed method is well-motivated and interesting.

We have provided individual responses to each of your reviews in the comments but we would also like to highlight that we have now uploaded a new version of the paper based on your feedback.

All changes in the text have been highlighted in blue for clarity and includes:

- Notation and clarifications
- Additional related work
- Details on computational complexity
- Added baselines in the MNIST example

Thanks again for your work, and if you have further questions please let us know!

---

### Meta-Review · Area_Chair_ymH2 · 2022-08-21

**Recommendation:** Accept
**Confidence:** Less certain

**Metareview:**

This work suggests that in cases where data is sensitive it might be easier to gain access to pre-trained models instead of to the data used for training them. However, since these models were trained on different distributions, their prediction may be better/worse depending on whether the point of interest in in the support of the distribution they trained on. Hence, the setting is a sort of learning from experts’ advice [1] where the best expert should be selected locally.

In this work it is assumed that each model (expert) is published together with some information about the distribution on which the model was trained. The assumption that such data may be provided is justified by the common practice of providing descriptive statistics of the data used in publications in the medical domain, typically in Table 1 of such papers.

Several related problems have been studied before this work. One of the main criticism reviewers had about this work was the incomplete positioning of this work in relation to these earlier studies, especially in the first version of this work submitted for review. The clarifications were given by the authors in the rebuttal. Some differences between this work in prior art might be since some earlier studies tried to provide theoretical guarantees which forced the use of stronger (and explicitly) assumptions. It may be that the current work did not have to make such assumptions since it does not contain theoretical analysis.

The term “synthetic” here is used as a nod to synthetic control. The idea is that both in synthetic control and here a convex combination of weak models is used. However, this is true for almost any ensemble model (bagging, random forest, adaboost…). Moreover, in synthetic control the weighting is fixed (global) as opposed to the main selling point of this work which is the local weighting.

A key assumption made in this work is that a model will be confident in its predictions on regions that are in the support of the dataset used for training it. This is stated, for example, in lines 184-185. This assumption is not always correct since the decision boundary of a model could be a region of high density on which the model is not confident about. Another case in which this assumption might fail is when the information about the distribution used for training, I_m, fails to provide good description. For example, if the data contains two well separated clusters and the information in I_m is the mean and the variance then the samples generated are likely to be from the area of low density between the two clusters on which the model might have low confidence.

The solution provided is based on intuition and is not equipped with theoretical support. The experiments show encouraging results, but they have their own limitations. For example, in the MNIST experiment, what is the information about the underlying distribution provided to the algorithm? The reviewers made several comments about the empirical evaluation and the authors discussed this at length in their response explaining that since this is a new problem domain, there are no standard benchmarks available.

Overall, this work studies an interesting problem and presents novel ideas. However, these ideas are not fully analyzed since there is no theoretical analysis and the empirical evaluation has its own limitations. It does seem that this work may contribute to problems such as medication dosing demonstrated in Section 4.3 but if this is the main contribution then it is not clear the NeurIPS is the right venue for such work.

This puts this work as a borderline case for NeurIPS: it does present new problem and some novel ideas; however, the analysis has many loose ends.


[1] Cesa-Bianchi, Nicolo, et al. "How to use expert advice." Journal of the ACM (JACM) 44.3 (1997): 427-485.


**Award:**

No

---

### Decision · Program_Chairs · 2022-09-14

Accept